# 3D Printing Custom Bioactive and Absorbable Surgical Screws, Pins, and Bone Plates for Localized Drug Delivery

**DOI:** 10.3390/jfb10020017

**Published:** 2019-04-01

**Authors:** Karthik Tappa, Udayabhanu Jammalamadaka, Jeffery A. Weisman, David H. Ballard, Dallas D. Wolford, Cecilia Pascual-Garrido, Larry M. Wolford, Pamela K. Woodard, David K. Mills

**Affiliations:** 1Mallinckrodt Institute of Radiology, Washington University School of Medicine, St. Louis, MO 63110, USA; ujammalamadaka@wustl.edu (U.J.); davidballard@wustl.edu (D.H.B.); woodardp@wustl.edu (P.K.W.); 2University of Illinois at Chicago Occupational Medicine, Chicago, IL 60612, USA; jeffery.weisman@gmail.com; 3Department of Surgery, UT Southwestern Medical Center; University of Texas Southwestern, Dallas, TX 75390, USA; dallaswolford@gmail.com; 4Adult Reconstruction-Adolescent and Young Adult Hip Service, Washington University Orthopedics, St Louis, MO 63110, USA; cpascualgarrido@wustl.edu; 5Department of Oral and Maxillofacial Surgery, Texas A&M University College of Dentistry and Baylor University Medical Center, Dallas, TX 75246, USA; lwolford@drlarrywolford.com; 6Department of Biomedical Engineering, Louisiana Tech University, 305 Wisteria Street, Ruston, LA 71272, USA; dkmills@latech.edu

**Keywords:** additive manufacturing, orthopedic fixation devices, antibiotics, chemotherapeutics, drug delivery implants, biodegradable, polylactic acid (PLA), 3D printing

## Abstract

Additive manufacturing has great potential for personalized medicine in osseous fixation surgery, including maxillofacial and orthopedic applications. The purpose of this study was to demonstrate 3D printing methods for the fabrication of patient-specific fixation implants that allow for localized drug delivery. 3D printing was used to fabricate gentamicin (GS) and methotrexate (MTX)-loaded fixation devices, including screws, pins, and bone plates. Scaffolds with different infill ratios of polylactic acid (PLA), both without drugs and impregnated with GS and MTX, were printed into cylindrical and rectangular-shaped constructs for compressive and flexural strength mechanical testing, respectively. Bland PLA constructs showed significantly higher flexural strength when printed in a Y axis at 100% infill compared to other axes and infill ratios; however, there was no significant difference in flexural strength between other axes and infill ratios. GS and MTX-impregnated constructs had significantly lower flexural and compressive strength as compared to the bland PLA constructs. GS-impregnated implants demonstrated bacterial inhibition in plate cultures. Similarly, MTX-impregnated implants demonstrated a cytotoxic effect in osteosarcoma assays. This proof of concept work shows the potential of developing 3D printed screws and plating materials with the requisite mechanical properties and orientations. Drug-impregnated implants were technically successful and had an anti-bacterial and chemotherapeutic effect, but drug addition significantly decreased the flexural and compressive strengths of the custom implants.

## 1. Introduction

The success of an osseous implant depends on many factors, including biocompatibility, osteointegration, good mechanical strength, resistance to corrosion, and aseptic loosening [1]. An ideal osseous implant should also have an elastic modulus similar to that of bone at the site of implantation. Different types of bones have different biomechanical properties. Human cortical bone has a compression strength ranging from 90 to 200 MPa [2], while for cancellous bone it ranges from 0.2 to 10.44 MPa [3]. Many commercial metallic osseous implants have limited ability for customization according to the site of implantation. Aseptic loosening, due to the formation of biofilms on the surface of implants, is one of the common causes for implant failure and often requires secondary or revision surgery to treat [4]. Additionally, conventional commercial orthopedic implants or orthopedic hardware do not have the ability to deliver drugs at the site of implantation to fight infection or cancerous cells.

Recent advances in additive manufacturing and biomaterials have shown tremendous potential for serving as drug-eluting implants. Since its first preliminary report in 1970 [5], biodegradable polymers have proven their efficacy as excellent bone implant materials [6]. They provide the required mechanical support at the site of implantation and resorb over time, transferring the load to newly formed tissue and avoiding the need for revision surgery for hardware removal [7]. Biopolymers come in various forms and can be easily molded into any complex geometries. Currently, many osseous fixation devices including screws, pins, plates, and rods are commercially made using biopolymers and are used for various dental and orthopedic applications. Since these polymers are resorbable, they can also act as drug carriers for localized drug delivery. Additive manufacturing or 3D printing technology provides greater customizability, speed, and accuracy. Due to these advantages and the capability of 3D printers to use biopolymers as their raw materials, the use of this technology in medical and pharmaceutical fields has increased tremendously. Over the last decade, fabricating implants [8], biomedical devices [9], and prostheses highly personalized to patients’ requirements [10] was made possible with this rapid prototyping technology. 

Using 3D printing technology, custom biodegradable polymers can be engineered into different shapes and compositions with specific degradation times. The mechanical strength of the 3D printed construct can be customized by altering the amount of polymer filling the interior of the constructs (infill). Additionally, the possibility of using these implants as localized drug delivery systems makes them an attractive potential tool for future orthopedic and maxillofacial applications. The purpose of this study is to demonstrate the feasibility of using 3D printing technology to fabricate biodegradable 3D printed screws, plating, and other osteofixators with customizable and optimized mechanical properties and drug impregnating capabilities. Biodegradable polymer, polylactic acid (PLA), will be used for carrying antibiotic (gentamicin sulfate) and chemotherapeutic (methotrexate) drugs. Various PLA constructs, with and without drugs, will be 3D printed in different geometries of orthopedic hardware. The mechanical properties of the polymer will be assessed using compression and flexural strength evaluation. The retention of antibacterial properties of the antibiotic constructs will be evaluated against *E. coli* and the chemotherapeutic activity against osteosarcoma cells.

## 2. Materials and Methods 

### 2.1. Materials

ExtrusionBot filament extruder (ExtrusionBot, LLC; Phoenix, AZ, USA) and a MakerBot replicator 3D printer (MakerBot; Brooklyn, NY, USA) were used for 3D printing. For modeling 3D constructs, Solidworks 2015 (Dassault Systems, MA, USA) was used. For bacterial culture, 100 mm Mueller Hinton agar plates were purchased from Fischer Scientific (Hampton, NH) and Escherichia coli ATCC 11,775 Vitroids 1000 CFU were from Sigma Aldrich (St. Louis, MO, USA). Methotrexate (MTX) and gentamicin sulfate (GS) were ordered from Sigma Aldrich (St. Louis, MO, USA). PLA pellets used for extruding filaments were obtained from Push Plastic (Springdale, AR, USA), KJLC 705 silicone oil used for coating pellets was purchased from Kurt J. Lesker Company (Jefferson Hills, PA, USA).

### 2.2. Methods

#### 2.2.1. Fabricating Drug Loaded Scaffolds

To impregnate printing materials with drugs, we used a previously-described oil coating method to coat pellets with the drugs [11]. These coated pellets were extruded, using ExtrusionBot filament extruder, at 170 °C into filaments of 1.75 mm diameter. These filaments were then used in the 3D printer to fabricate required constructs. All 3D CAD models were designed using Solidworks 2015 software (Dassault Systems, MA, USA). Makerbot 5th generation desktop 3D printer (MakerBot, Brooklyn, NY, USA) was used to fabricate the constructs. The print-head temperature was maintained at 215 °C at a filament feed rate of 20–23 mm/s and a print-head speed of 12–8 mm/s. 

#### 2.2.2. Mechanical Evaluation

We aimed to customize the properties of the implants by changing the printing parameters to make them available for a wide range of osteofixation applications, specifically to optimize properties such as hardness, elasticity, yield stress, wearability and degradation time. Scaffolds with different infill ratios, different orientations, and which were drug loaded, as shown in Figure 1; Figure 2, were printed. These constructs were subject to compression and flexural testing.

Compression cylinders with dimensions 6 × 12 mm and flexural bars of dimensions 75 × 10 × 4 mm^3^ were 3D printed for evaluation. For testing the mechanical properties, both compression and flexural testing were performed using an Admet 2600 Dual Column Bench Top Universal Testing Machine (Norwood, MA, USA). For data acquisition and analysis, MTESTQuattro software (Version 4.0, ADMET, Norwood, MA, USA) was used. For both tests, ASTM F451-99a (characterization of mechanical properties of bioresorbable scaffolds) guidelines were followed [12]. Load capacity of 1 kN was laid on the scaffolds at a rate of 1 mm/min. For flexural testing the three-point bending method was followed.

#### 2.2.3. Antibacterial and Chemotherapeutic Properties

To assess the bacterial activity of GS, zone of inhibition studies were conducted on standard Muller Hinton Agar Plates (Fischer Scientific, Hampton, NH, USA) using *E. coli*. For repeatability, vitroids of this commercial bacterial strain was used (Sigma Aldrich, Saint Louis, MO, USA). These vitroids were rehydrated and inoculated into agar plates as per the procedures suggested in the manual. Plain PLA pellets, coated PLA pellets, extruded PLA filaments, GS loaded PLA filaments, beads, and constructs were tested. Discs of 5 × 1 mm were 3D printed using PLA–GS and PLA–MTX filaments and were evaluated. To make sure there was no contamination among agar plates, a blank plate was cultured. An agar plate with just the bacteria acting as negative control and another with a standard GS disc as positive control were also cultured. Under aseptic conditions, samples were inoculated along with the bacteria in the plates and incubated at 37 °C for 24 h. The diameter of inhibition zones was measured at three different points including the samples at the center of the zone and averaged.

#### 2.2.4. Statistical Analysis

All mechanical testing samples were studied in multiples of five to generate mean data. Five samples for each batch (infill ratio and drug concentrations) were evaluated. For comparing the means of the strengths among the different groups, one-way ANOVA test was performed at a significance level of 0.05. Post hoc analysis was performed to calculate the highest average among the groups. For evaluating the antibiotic activity, five agar plates for each composition were tested. For all the data, standard deviation was calculated and used in the graphs as error bars.

## 3. Results

PLA pellets were successfully coated with drugs GS and MTX using the oil coating method. Optimization of the amount of silicone oil used to coat the pellets was challenging in early iterations. Excess oil resulted in clumping of pellets together and clogging of the extruder during the extrusion process. However, low amounts of oil would only allow pellets to hold small proportions of drugs on to their surface. For a batch of 20 g pellets, 10–15 μL of silicone oil was found to yield the best results. Filaments of 1.75 mm diameter were successfully extruded from coated pellets using the ExtrusionBot extruder (ExtrusionBot LLC, Phoenix, AZ, USA) at 170 °C. These filaments were used in the MakerBot 3D printer and scaffolds were printed. Figure 3 and Figure 4 show images of scaffolds 3D printed using custom extruded drug-impregnated filaments.

### 3.1. Mechanical Properties

The flexural strengths of different 3D printed control PLA scaffolds with different infill ratios and orientations are summarized in Figure 5**.**

There was no significant difference seen between 25%, 50%, and 75% infill samples, and the 100% infill samples had the highest flexural strengths. Among different orientations, samples printed along the Y axis showed high flexural strengths. Samples printed along the X axis had 8% less resistance and along the Z axis had 26% less flexural strength when compared to Y axis prints. This might be due to the alignment of filament layers along the length of the specimens, whereas, in other orientations, filaments were sintered with each other perpendicular to the length of the specimen. 

The graph in Figure 6 shows the flexural strengths of PLA samples printed with and without drugs. Extruded control PLA prints had a mean flexural strength of 78 MPa. Samples printed with GS and MTX showed a 14.4% and 17.1% decrease in strength, respectively.

Unlike flexural strengths, compression strengths of the samples printed with different infill ratios showed a significant difference in their mean values. The graph in Figure 7 shows the stress–strain curves of PLA constructs, and Figure 8 shows the compression strengths of samples printed with different infill ratios and in various orientations. Samples printed at 100% infill ratio showed the highest compression resistance among other infills. There was no significant difference in means of compressive strengths printed at different orientations.

The graph in Figure 9 shows the compression strengths of PLA samples with different drugs. The control PLA scaffolds had compression pressure of 67.66 MPa. Scaffolds with GS and MTX had 48% and 42% reduction in compression strength, respectively.

### 3.2. Antibacterial and Chemotherapeutic Properties

Figure 10 shows the zone of inhibition for GS-coated pellets, filaments, and 3D printed discs. PLA pellets coated with 2.5 wt. % GS showed an average of 29.04 mm diameter zone of inhibition. Control PLA and Poly (methyl methacrylate) (PMMA) filaments, as shown in Figure 10B, did not show any kill zones. Clear demarcating inhibition zones were seen for PLA and PMMA filaments loaded with GS. The mean zone of inhibition for 2.5 wt. % PLA–GS was 23.13 mm and for 2.5 wt. % PMMA–GS was 22.58 mm. ANOVA analysis showed no significant difference between the two means. 

Similarly, the mean zone of inhibition diameters of 3D printed discs and hand mold PMMA discs were 21.36 mm and 22.02 mm, respectively. Figure 11 shows the diameters of zones of inhibitions for various constructs. ANOVA analysis showed no significant difference in the mean values of both groups.

The 3D printed screws, plates, and pins also showed a clear demarcating zone of inhibition. Figure 12 and Figure 13 show the inhibition zones for PLA–GS catheter incubated with *E. coli* on comparison with control screws and plates.

The XTT assay performed on osteosarcoma cell lines showed lower cell viability in wells containing PLA–MTX when compared with the control PLA groups, as shown in Figure 14, indicating the presence of chemotherapeutic properties in the 3D printed constructs.

## 4. Discussion

The drug-impregnated implants investigated in this study were successfully fabricated using biodegradable polymer–PLA and additive manufacturing technology. Control 3D printed constructs demonstrated adequate mechanical properties with degrees of customization when adjusting the infill ratio and axis of printing. Therefore, implants with varied Young’s modulus can be fabricated and customized according to the site of implantation. There was a significant reduction in mechanical strengths of the PLA constructs with the addition of drug-containing compounds. Increasing the strengths of the constructs using nanoparticle technology or by using other additive polymers should be an area of further investigation. The type of infill pattern is an important factor determining the mechanical strength of the constructs. Ali Nadernezhad et al., have shown that 3D constructs with honeycomb-shaped infill patterns had more isotropic mechanical properties than other infill patterns [13]. All the constructs used for the evaluation were 3D printed in honeycomb patterns of infill. GS-impregnated implants successfully deterred bacterial growth, and MTX-impregnated implants showed a chemostatic effect on osteosarcoma cells. Assays performed on these two drugs imply that drugs can be successfully loaded in the biopolymer and can potentially act as drug delivery systems. 

The non-3D printed fracture fixation devices fabricated from different lactic acid-based polymers, such as poly L-lactic acid (PLLA), poly lactic glycolic acid (PLGA), and poly caprolactone (PCL), have compression strength values of 80–500 MPa, 40–55 MPa, and 20–40 MPa, respectively [14]. While the 3D printed PLA constructs had compression strength of 67.66 MPa, indicating their feasibility in orthopedic applications. The present study showed feasibility of 3D printing screws, plating, and other constructs impregnated with drugs, and profiled the mechanical properties effect of incorporating these drugs into the 3D printed constructs. The drug impregnating capabilities are similar to prior in vitro studies that incorporated antibiotics and chemotherapeutics into 3D printed constructs [8,9,11,15,16,17]. Several studies have shown antibiotics and chemotherapeutics incorporated into 3D printed filaments, catheters, and meshes [9,11,15]. Additional drug-incorporating studies into 3D printed constructs have shown the capability to have implants impregnated with contrast materials [16,17], hormones [8], and act as a scaffold for tissue cultures [18]. 

Fabricating osseous fixation devices that can elute drugs using 3D printing techniques have not been previously described to the best of our knowledge. In the present study, 3D printing using the fused deposition modeling technique was used to avoid elaborate post-sintering processes (heating at high temperatures or polymerization using toxic monomers) and to allow biopolymers to successfully carry drugs without effecting their intended effects. Our study demonstrates broad feasibility that PLA can be patient-specific and serve as a potential implant used for localized drug delivery. 

3D printed constructs that are implanted in patients or animals are often post-processed to increase their material properties either by sintering at high temperatures or by surface polymerization [19]. Khalyfa et al. [13] used a powder-based 3D printer to fabricate patient-specific implants from calcium phosphate cement. These printed objects were post-processed either at higher temperatures or using liquid monomers to increase their mechanical properties. Strobel et al. [14] used a binder jetting technique to 3D print calcium phosphate constructs and sintered at 1200 °C. These constructs were soaked in the basal medium along with BMP-2 to enhance the bone formability of the implants in rats. Similarly, Martinez-Vazquez et al. [15] used a material extrusion technique to 3D print silicon-doped hydroxyapatite/gelatin scaffolds loaded with vancomycin for orthopedic drug delivery applications. The antibacterial property was successfully evaluated by bacterial inhibition studies. However, these 3D printed constructs were soaked in glutaraldehyde for 10 min for polymerization and then either air-dried or lyophilized at 80 °C for structural integrity.

Systemic drug delivery has the risk of an adverse systemic effect that may be circumvented with personalized drug-impregnated 3D printed implants. Localized drug delivery can prevent these risks and deliver drugs in sufficient concentrations to the targeted site. Most commercially available osseous implants are metallic and lack drug delivering capabilities. Efforts of prior studies have therefore focused on formulating biodegradable orthopedic materials that have similar mechanical properties of bone and can be impregnated with drugs [20]. Commercially available osseous fixation devices, such as metallic and ceramic implants, are often excessively stiff and do not have the ability to deliver drugs. They are at a permanent risk of infection and may require removal or secondary surgery [21]. Furthermore, some conventional implants cannot be tailored to patient-specific customization. The proposed implants can be 3D printed into any required geometry (specific to the patient) and can deliver a wide range of drugs to the intended site avoiding the side effects of systemic drug delivery. Different from hardware devices commonly used in orthopedics, 3D printed structures can provide additional biological local treatment to severe orthopedic conditions, such as delivery of antibiotics for treatment of osteomyelitis or chemotherapy drugs for the treatment of soft or bone tissue diseases such as osteosarcoma [22]. 

As this is a proof of concept study, this research was limited to using only one kind of polymer (PLA) and two different drugs (GS and MTX). The 3D printing technique used herein is highly versatile and can be easily molded to the inclusion of alternative polymers (or composite polymers) and varied drugs or other agents depending upon the application. Another limitation of this study is that only a single drug concentration was used for both GS and MTX. Future studies should aim to correlate the effect of the concentration of drugs on mechanical properties of the polymer.

Future studies are needed to demonstrate efficacy of drug-impregnated 3D printed implants in animals and humans, such as the drug-impregnated screws and plates in the present study. A number of regulatory considerations are important to consider regarding the 3D print implants role as a device [19,23,24]. The US Food and Drug Administration have published a number of statements regarding their stance on regulating 3D printed constructs in medicine [23,24]. The Radiological Society of North American 3D Printing Special Interest Group has also published guidelines for medical 3D printing [25]. Following these guidelines will become increasingly relevant in the US with upcoming category III Current Procedural Terminology for 3D printed anatomic models and surgical guides codes beginning in July 2019 [26]. 

Devices can have varying concentrations of drugs as well as only specific portions that contain a drug for localized delivery. Additionally, hybrid fabrication can be used to create devices that are partially absorbable or adsorb on a custom timeline. In this research, osseous fixation devices were successfully 3D printed using fused deposition modeling techniques. We have demonstrated that the geometry and mechanical strength of the implants can be altered in accordance with patient-specific requirements and site of implantation. Additionally, antibiotics (GS) and chemotherapeutics (MTX) were successfully loaded into the implant material for localized drug delivery. These preliminary findings of drug-impregnated osseous fixation implants require future validation in animal models to demonstrate both feasibility and if they confer an advantage over traditional implants.

## Figures and Tables

**Figure 1 jfb-10-00017-f001:**
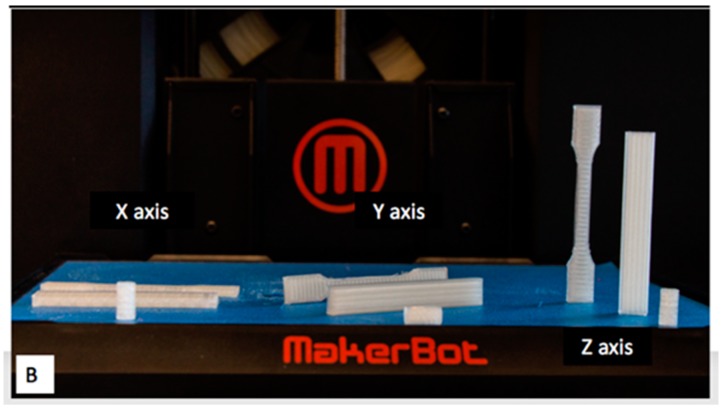
3D Printed polylactic acid (PLA) constructs (compression cylinders, flexural bars and dog-bone shape) in different orientations.

**Figure 2 jfb-10-00017-f002:**
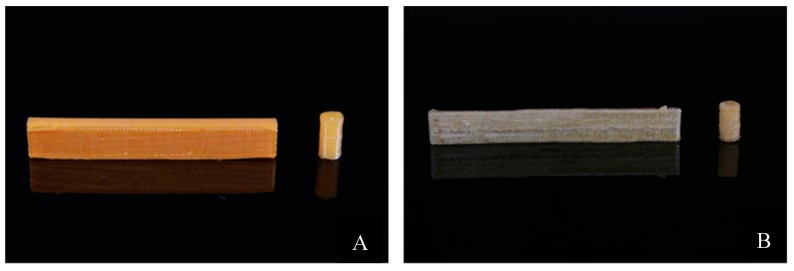
3D printed flexural bars and compression cylinders. (**A**) Methotrexate (MTX)–PLA mechanical testing samples, (**B**) gentamicin sulfate (GS)–PLA mechanical testing samples.

**Figure 3 jfb-10-00017-f003:**
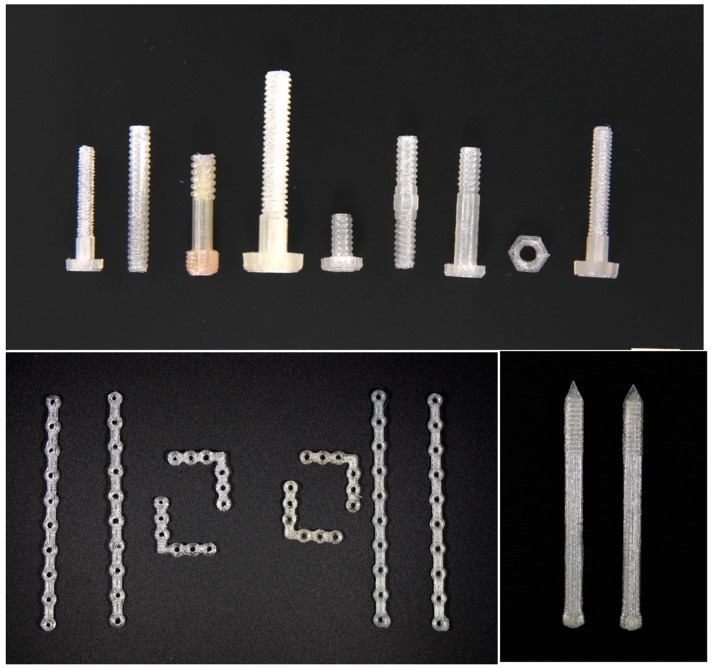
3D Printed PLA orthopedic screws, pins, and plates.

**Figure 4 jfb-10-00017-f004:**
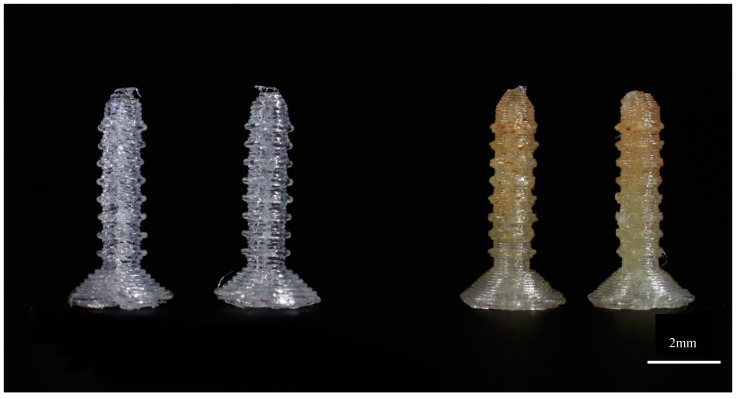
3D printed standard 4 mm screws using control PLA and PLA–GS.

**Figure 5 jfb-10-00017-f005:**
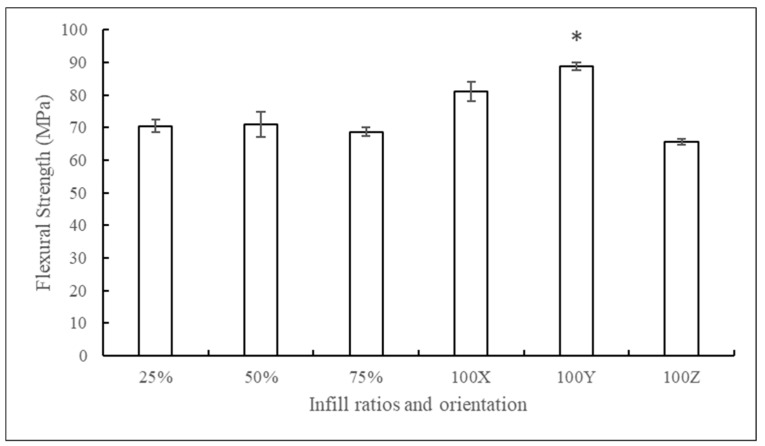
Flexural strengths of PLA constructs printed in different infill ratios and orientations (mean ± SD, n = 5). * for p < 0.05.

**Figure 6 jfb-10-00017-f006:**
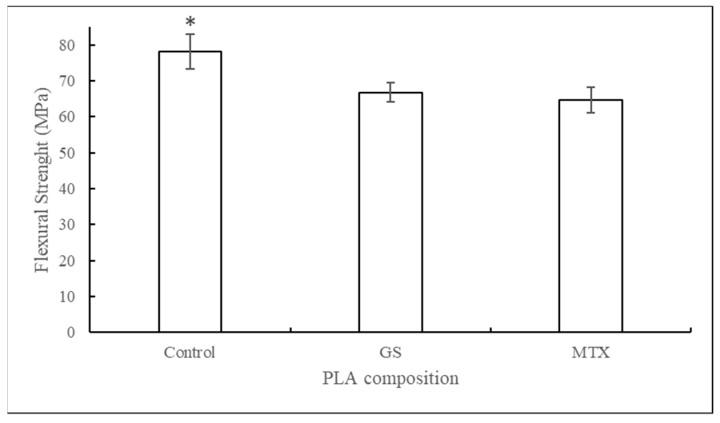
Flexural strengths of PLA constructs printed with control, GS, and MTX (mean ± SD, n = 5). * for p < 0.05.

**Figure 7 jfb-10-00017-f007:**
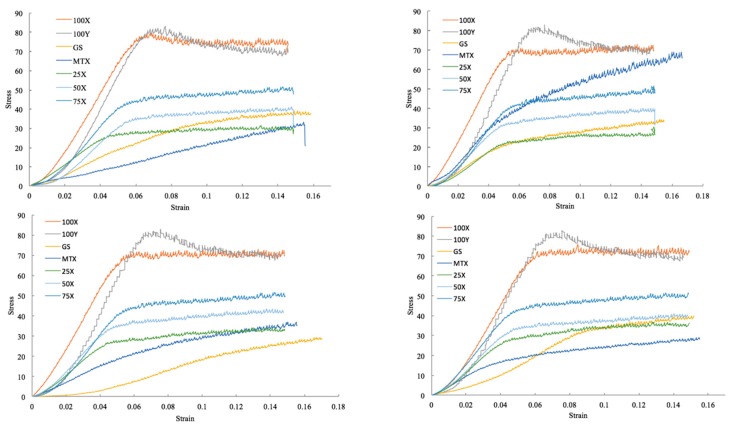
Stress–strain curves for different trials of PLA constructs.

**Figure 8 jfb-10-00017-f008:**
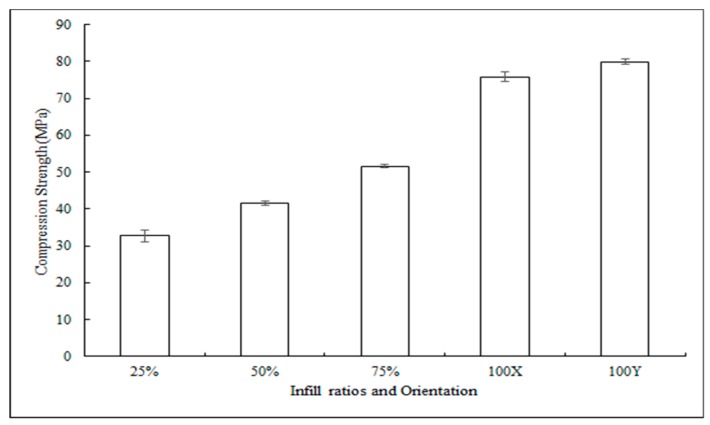
Compressive strengths of samples printed with different infill ratios and in various orientation constructs (mean ± SD, n = 5). * for p < 0.05.

**Figure 9 jfb-10-00017-f009:**
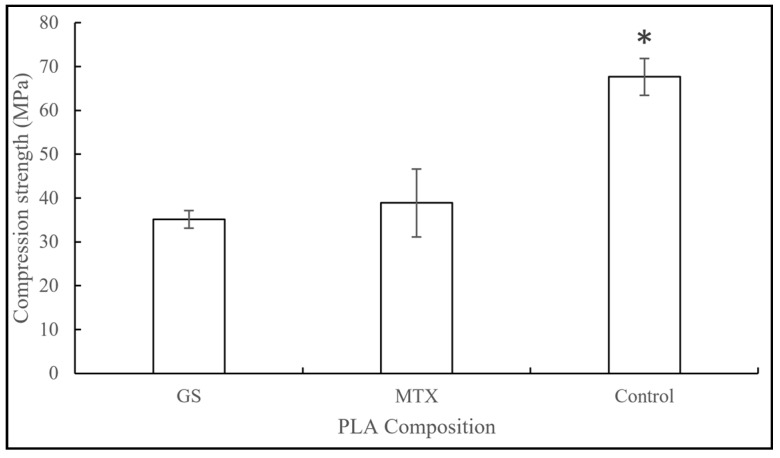
Compressive strengths of control, GS, and MTX–PLA constructs (mean ± SD, n = 5). * for p < 0.05.

**Figure 10 jfb-10-00017-f010:**
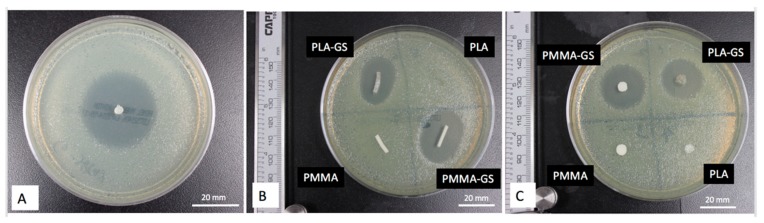
Zone of Inhibitions for *E. coli* cultures. (**A**) PLA–GS pellet, (**B**) PLA and PMMA filaments with and without GS, (**C**) PLA and PMMA discs with and without GS.

**Figure 11 jfb-10-00017-f011:**
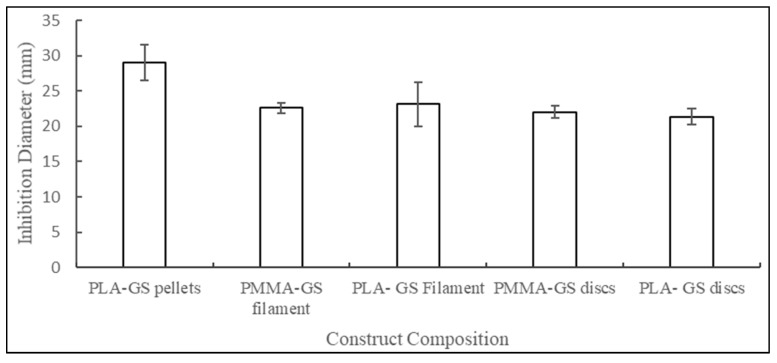
Inhibition diameters for *E. coli* cultures against PLA–GS pellet; PLA and PMMA filaments; PLA and PMMA.

**Figure 12 jfb-10-00017-f012:**
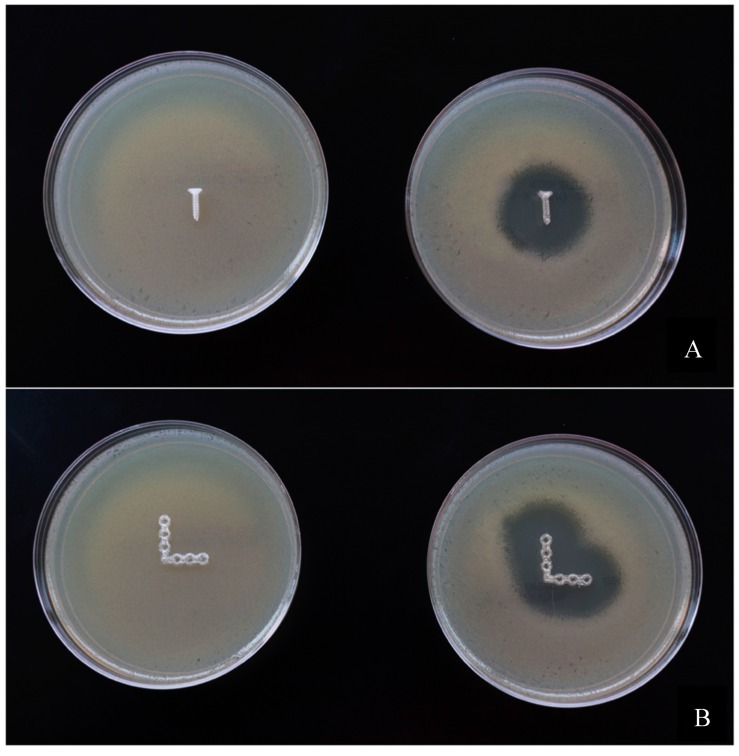
Bacterial growth Inhibition of *E. coli* on Mueller–Hinton agar plates. (**A**) 4 mm Screws, (**B**) bone plates.

**Figure 13 jfb-10-00017-f013:**
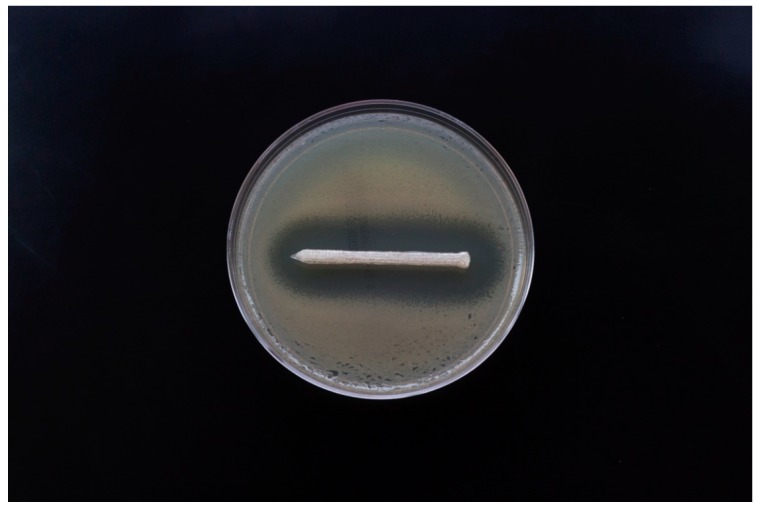
Zone of inhibition of 3D printed PLA–GS pin.

**Figure 14 jfb-10-00017-f014:**
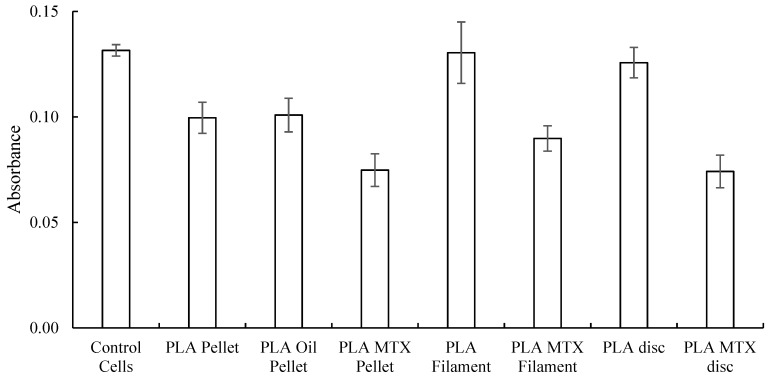
Absorbance values of XTT assay performed on osteosarcoma cells with different PLA scaffolds enhanced with MTX (mean ± SD, n = 6).

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
