# Peer review of "3D Printing Custom Bioactive and Absorbable Surgical Screws, Pins, and Bone Plates for Localized Drug Delivery"

_jfb, 2019, doi:10.3390/jfb10020017_

Round 1

Reviewer 1 Report

Thank you for submitting your work. Here are some observations/questions that need to be considered by the authors:

-          Page 4 – the type of infill (geometry) is also important and can impact the flexural strengths; please explain further and/or add some experiments; what are the 3D printing parameters used?

-          The amount of drug oil that remains on the filament after extrusion is very important; also the amount of drug oil that still remains after practically the second extrusion when printing the orthopaedic screws, etc. is more important; hence this analysis of flexural strengths should be done in correlation with the amount of retained drug within the materials; this an analysis of both results in 2.2 and 2.3 should be made

-          Section 2.3 – there are no characteristics given for the 3D printed discs and screws; how many printed discs/screws were printed and tested? Is there a significant difference between them? There is a mention of 5 samples in section 2.1.4, but it is unclear; What was the extrusion temperature? Is it relevant (and it is)? Etc. These questions are quite important

-          Discussions section – the authors claim that 3D printed devices can be used in orthopaedics from the flexural strengths point of view; is this true from other points of view? Could you give some example of applications?

-          I suggest you move the section 4 (that has also incorrectly number subchaptered) before the Discussion and results sections

Author Response

Please find the uploaded word document named "Reviewers comments reply" containing the authors response to the reviewers comments/suggestions.

Reviewer 2 Report

Interesting work needing some improvements:

1. This recent paper should be cited in the Introduction to underline the interest in antiseptic biomaterials:

Mesoporous bioactive glasses: promising platforms for antibacterial strategies. Acta Biomaterialia 2018;81:1-19.

2. Figure 7: ‘Mpa’ should be ‘MPa’ on the y-axis.

3. The results about the inhibition zones should be reported as avg +/- SD in a proper table.   

4. The authors use the term ‘bioactivity’ to indicate the antiseptic properties of the samples. This is incorrect, as bioactivity usually refers to another thing (i.e., apatite-forming ability and bone-bonding) in the biomaterials community. Please replace and use the expression ‘antibacterial properties’ throughout the manuscript.

Author Response

Please find the attached word document named "Reviewer2 Comments Reply" containing authors reply to the reviewers comments/suggestions.

Round 2

Reviewer 1 Report

Thank you for considering my comments.